https://doi.org/10.1038/s42003-021-01708-y | OPEN
# Insulin-like 3 affects zebrafish spermatogenic cells directly and via Sertoli cells

Diego Crespo[1,7,10], Luiz H. C. Assis[1,10], Yu Ting Zhang[2,8], Diego Safian[1,9], Tomasz Furmanek[3], Kai Ove Skaftnesmo[3], Birgitta Norberg [4], Wei Ge[5], Yung-Ching Choi[5], Marjo J. den Broeder[6], Juliette Legler[6], Jan Bogerd[1] & Rüdiger W. Schulz [1,3 ✉]

Pituitary hormones can use local signaling molecules to regulate target tissue functions. In adult zebrafish testes, follicle-stimulating hormone (Fsh) strongly increases the production of insulin-like 3 (Insl3), a Leydig cell-derived growth factor found in all vertebrates. Little information is available regarding Insl3 function in adult spermatogenesis. The Insl3 receptors Rxfp2a and 2b were expressed by type A spermatogonia and Sertoli and myoid cells, respectively, in zebrafish testis tissue. Loss of *insl3* increased germ cell apoptosis in males starting at 9 months of age, but spermatogenesis appeared normal in fully fertile, younger adults. Insl3 changed the expression of 409 testicular genes. Among others, retinoic acid (RA) signaling was up- and peroxisome proliferator-activated receptor gamma (Pparg) signaling was down-regulated. Follow-up studies showed that RA and Pparg signaling mediated Insl3 effects, resulting in the increased production of differentiating spermatogonia. This suggests that Insl3 recruits two locally active nuclear receptor pathways to implement pituitary (Fsh) stimulation of spermatogenesis.

[1] Reproductive Biology Group, Division Developmental Biology, Department of Biology, Science Faculty, Utrecht University, Utrecht, The Netherlands. [2] State Key Laboratory of Marine Environmental Science, College of Ocean and Earth Sciences, Xiamen University, Fujian, PR China. [3] Research Group Reproduction and Developmental Biology, Institute of Marine Research, Bergen, Norway. [4] Institute of Marine Research, Austevoll Research Station, Storebø, Norway. [5] Center of Reproduction, Development and Aging (CRDA), Faculty of Health Sciences, University of Macau, Taipa, Macau, China. [6] Division of Toxicology, Institute for Risk Assessment Sciences, Faculty of Veterinary Medicine, Utrecht University, Utrecht, The Netherlands. [7] Present address: Research Group Reproduction and Developmental Biology, Institute of Marine Research, Bergen, Norway. [8] Present address: Institute of Oceanography, Minjiang University, Fuzhou, PR China. [9] Present address: Experimental Zoology Group and Aquaculture and Fisheries Group, Department of Animal Science, Wageningen University, Wageningen, The Netherlands. [10] These authors contributed equally: Diego Crespo, Luiz H. C. Assis. ✉email: r.w.schulz@uu.nl

The relaxin-like peptides represent a family of peptide hormones, which have evolved in both vertebrates and invertebrates showing a rigid peptide scaffold involving multiple cysteine bridges, common also to insulin and the insulin-like growth factors[1,2]. Insulin-like 3 (INSL3) is a member of this family and exerts biological activity via its receptor RXFP2[3–5]. Loss of INSL3 or of its receptor RXFP2 results in cryptorchidism in mice, reflecting the importance of INSL3 for the proper testicular descent into the scrotum during the fetal life of most mammals[6–9]. INSL3 is preferentially expressed in gonadal tissues, and at particularly high levels in the testicular Leydig cells[10,11]. INSL3 production continues during postnatal life and biological activities included the reduction of germ cell loss via apoptosis in rat and boar testis[12–15], or increased testosterone production by primary mouse Leydig cell cultures[16]. RXFP2 is also expressed outside the reproductive system, for example in osteoblasts, where INSL3 triggered osteocalcin release, in turn activating via its receptor GPRC6A Leydig cell androgen production, independent of luteinizing hormone (LH)[17].

INSL3/Insl3 has also been studied in non-mammalian vertebrates, including fish[18]. Since a descent of the testes during fetal life only occurs in mammals, non-mammalian vertebrates are excellently suited to study other biological activities of Insl3. Prominent expression of the gene encoding Insl3 (insl3) by Leydig cells is a conserved feature also found in teleost fish, for example in the zebrafish, Danio rerio[19]. In this species, insl3 transcript levels increased strongly in response to follicle-stimulating hormone (Fsh) but not to Lh[20,21]. In fish, Leydig cells also express the receptor for Fsh, rendering it a potent steroidogenic gonadotropin[21,22]. Follow-up studies showed that both, human INSL3 and zebrafish Insl3, stimulated the differentiating proliferation of type A undifferentiated (Aund) spermatogonia, while no direct effect was found on testicular androgen production in zebrafish[20,23]. These studies suggest that Fsh-induced stimulation of spermatogenesis is mediated, at least in part, by Insl3[20]. However, the mechanism(s) by which Insl3 promotes spermatogonia proliferation and differentiation remain unknown.

While studies using Insl3 in primary zebrafish testis tissue culture experiments were informative[20,23], it was not known which receptor(s) mediate these effects. Our first aim was to identify the relevant testicular Insl3 receptor(s) from candidate Rxfp receptors previously shown to be highly expressed in zebrafish testis[24]. Moreover, in order to learn more about the downstream effects of Insl3, we characterized the testicular phenotype after CRISPR/Cas9-induced loss of insl3 gene function, and we studied Insl3-induced changes in testicular gene expression by RNA sequencing (RNAseq). Pparg (peroxisome proliferator-activated receptor gamma) signaling was retrieved from this data set, so that we examined the effect of the loss of pparg gene function on the germ cell composition in zebrafish. Also, retinoic acid (RA) signaling was retrieved and triggered follow-up studies. The Insl3-mediated up-regulation of RA signaling as well as the down-regulation of Pparg signaling both promoted the production of differentiating spermatogonia, identifying two nuclear receptor pathways to mediate testicular growth factor signaling in response to a pituitary gonadotropin.

## Results

### Rxfp2a and Rxfp2b mediate Insl3 effects in the zebrafish testis.

4 of the 11 relaxin family peptide receptor genes (rxfps) in the zebrafish genome had highest homology to the mammalian Rxfp2 receptor and were also expressed in testis tissue[24]. We expressed each of these four receptors in HEK293T cells that were co-transfected with a construct harboring a cAMP-sensitive reporter gene that can be assessed using a colorimetric β-galactosidase assay. Zebrafish Insl3 increased intracellular cAMP levels in a dose-dependent manner for cells expressing Rxfp2a and Rxfp2b with $EC_{50}$ concentrations of 96.2 and 6.5 ng/mL, respectively (Fig. 1A). The Rxfp1 and the Rxfp2-like receptors both showed a lower level of maximum activity and were clearly less responsive ($EC_{50}$'s of 0.5 and 13.4 μg/mL, respectively) to zebrafish Insl3 (Fig. 1A).

Next, we analyzed rxfp2a and rxfp2b transcript levels in an RNAseq dataset that compared control, germ cell-depleted (following treatment with the cytostatic agent busulfan), and recovering testis tissue[25], to obtain information on the identity of receptor expressing cells. rxfp2a expression was enriched in germ cells, since its transcript levels were low in germ cell-depleted testes and increased to control levels during the recovery of spermatogenesis (Fig. 1B). rxfp2b expression, on the other hand, remained unchanged following germ cell depletion and subsequent recovery of spermatogenesis (Fig. 1B), suggesting that the rxfp2b transcript is mainly expressed by somatic cells in zebrafish testis.

To study their cellular expression in the adult zebrafish testis, transgenic rxfp2a:EGFP and rxfp2b:mCherry lines were studied. Confocal laser scanning microscopy of testis sections confirmed specific EGFP expression in germ cells, preferentially in type A spermatogonia (Fig. 1C), while mCherry expression was localized to somatic cells situated in the periphery of the spermatogenic tubules (Fig. 1D), Sertoli cells within the tubules, and myoid cells on the outside of the tubular wall. However, there were also Sertoli and myoid cells not showing the mCherry signal (Fig. 1D, white arrowheads).

### Genetic ablation of insl3 disturbs testis morphology but not fertility.

We generated an insl3 knockout line using CRISPR/Cas9 to investigate its role in the regulation of zebrafish spermatogenesis (Supplementary Fig. 1A). F0 founders had a deletion of 24 and an insertion of seven nucleotides resulting in a premature stop codon (Supplementary Fig. 1B). Furthermore, homozygous mutants (insl3^uua1, hereafter referred to as insl3^−/−) showed a strong reduction (at least 500-fold) of insl3 mRNA levels in the adult testis compared to wild-type males (Supplementary Fig. 1C).

Analysis of testis tissue from 3 and 6 months-old adult homozygous F3 insl3^−/− mutants did not show obvious defects in spermatogenesis and the fish showed normal fertility in both incrosses and outcrosses. However, in older adult males of 9 and 12 months of age, lack of insl3 resulted in effects on body and gonad weight. While body weight was only reduced at 9 months of age (Fig. 2A), a decrease in the gonado-somatic index (GSI) was found in 9 and 12 months-old adult mutants compared to wild-type siblings (Fig. 2B). For both parameters, but more clearly regarding body weight, a wider spread of the data was observed among the insl3 mutants (Fig. 2A, B). Morphological evaluation of mutant testis tissue at 9 months of age revealed an increased number of apoptotic germ cells (encircled by a yellow dashed line; Fig. 2C, lower panel). In these sections, germ cells were classified as apoptotic when showing shrinkage, leading to a loss of contact of the affected cell with its environment, and pyknosis/nuclear fragmentation. Testis morphology was more severely affected in 12 months-old insl3^−/− males, including abnormal cystic organization, germ cell-depleted areas and again a high incidence of germ cell apoptosis (Fig. 2C). Quantitative evaluation of spermatogenesis at 12 months of age showed relative smaller areas occupied by type B spermatogonia, spermatocytes and spermatids (Fig. 2D). The category "Others" (composed of (i) empty spaces that are not part of the lumen and are lined by

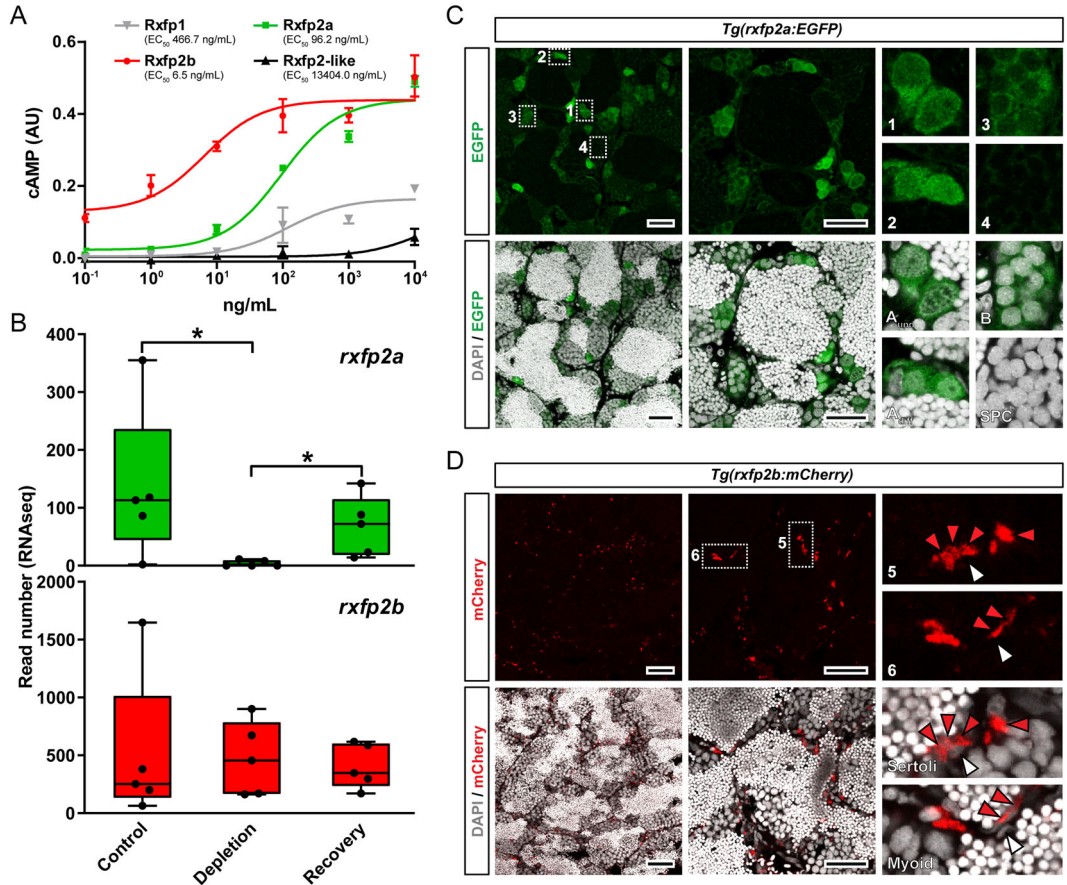

**Fig. 1 Rxfp2a and Rxfp2b mediate Insl3 action in the zebrafish testis. A** Effects of zebrafish Insl3 on four relaxin family peptide receptors (Rxfps) expressed in HEK293T cells transiently transfected with pcDNA3.1 and pCRE plasmids. Data are expressed as mean ± SEM ($N = 3$, technical replicates). Numbers in brackets indicate the $EC_{50}$ concentrations. AU, arbitrary units. **B** Expression levels of responsive Insl3 receptors in control, germ cell-depleted (by exposure to the cytostatic agent busulfan[51]), and testes with recovering (from busulfan) spermatogenesis, as described by Crespo et al.[25] (NCBI GEO data set GSE116611). Data are expressed as mean ± SEM ($N = 5$; *$p < 0.05$). **C**, **D** Localization of *rxfp2a:EGFP* (green; **C**) and *rxfp2b:mCherry* signal (red; **D**) in adult testis tissue. Confocal laser scanning microscopy analysis of whole-mount testes shows preferential expression of EGFP and mCherry in type A spermatogonia and Sertoli and myoid cells, respectively. DAPI counterstain is shown in gray. Representative germ cell types (insets 1–4) and Sertoli and myoid cells are shown (insets 5–6). $A_{und}$, type A undifferentiated spermatogonia; $A_{diff}$, type A differentiating spermatogonia; B, type B spermatogonia; SPC, spermatocytes. In **D**, red and white arrowheads indicate representative mCherry⁺ and mCherry⁻ somatic cells, respectively. Scale bars, 25 µm.

Sertoli and/or germ cells, (ii) Sertoli cell only areas, and (iii) apoptotic cells; see Supplementary Fig. 2) increased strongly in 12 months adult mutant testes (Fig. 2D). Despite the apparent defects on testis morphology, the area occupied by mature sperm was unaffected in adult *insl3*⁻/⁻ males (Fig. 2D). Also 12 months-old mutant males were able to induce spawning and to fertilize eggs from wild-type females.

Growth factor gene expression analyses in 12 months-old mutant males showed a consistent down-regulation of *igf3* and *amh*, but not of *gsdf*, a third, also Sertoli cell-derived growth factor (Fig. 2E and Supplementary Fig. 3A). Another gene expressed by Sertoli cells in the mammalian testis encodes the Gap junctional protein CX43, required for different aspects of the structural and functional integrity of Sertoli cells both, before and after puberty[26]. In *insl3*⁻/⁻ zebrafish, *cx43* transcript levels were slightly reduced in 9, and significantly reduced in 12 months-old mutants (Supplementary Fig. 3A). Both *gsdf* and *cx43* transcripts showed a pattern suggesting somatic expression (Supplementary Fig. 3B).

Of the genes involved in steroid production (*cyp17a1*, *hsd3b1*, and *star*) only the transcript level for the androgen producing enzyme *cyp17a1* was decreased (Fig. 2E), while *rxfp2a* and *rxfp2b* gene expression did not change in mutant testis tissue of one year

old fish (Fig. 2E). In 9 months-old mutants, on the other hand, when morphological changes were already visible but less clearly than 3 months later, *igf3* and *amh* transcript levels were not altered yet, while *cyp17a1* was 3-fold up-regulated (Supplementary Fig. 4).

**Insl3 acts as a germ cell survival factor**. Considering that genetic ablation of *insl3* increased the incidence of germ cell apoptosis, we sought to confirm the morphological observations by other approaches. First, we found that TUNEL-positive cells were significantly more frequent in *insl3*⁻/⁻ than in wild-type testes at 9 and in particular at 12 months of age (Fig. 3A, C). In 9 months-old *insl3*⁻/⁻ males, somatic cells in the germinal epithelium (potentially Sertoli cells, as suggested by the shape of the nuclei; white arrows) and some spermatogonia showed TUNEL-positive staining (Fig. 3A, white arrowheads, and B). However, in 12 months-old mutants, TUNEL-positive cells were mainly spermatocytes and spermatids, identified by the shape and size of their propidium iodide-stained nuclei (Fig. 3A, gray arrows and arrowheads, respectively, and B). Second, a significant up-regulation of the pro-apoptotic factor *casp9* and down-regulation of the anti-apoptotic factor *xiap* was found in 12 months-old, *insl3*⁻/⁻ males (Fig. 3D).

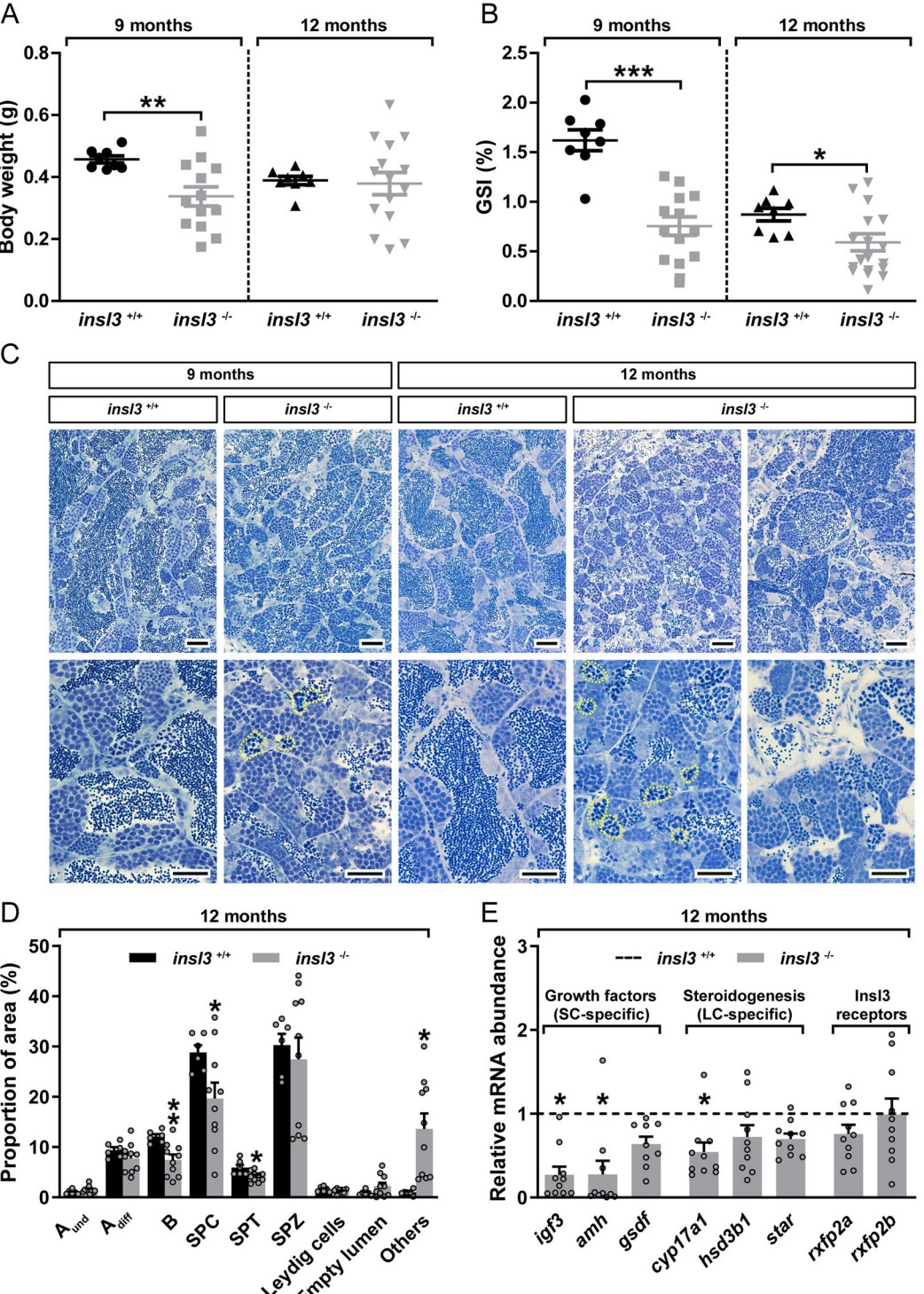

**Fig. 2 Genetic ablation of the *insl3* gene results in increased germ cell apoptosis in the zebrafish testis. A–C** Body weight (**A**), gonado-somatic indices (GSI; **B**) and testicular morphology (**C**) of wild-type (*insl3*$^{+/+}$) and *insl3* knockout (*insl3*$^{-/-}$) males 9 and 12 months post-fertilization. Data are mean ± SEM (*insl3*$^{+/+}$ and *insl3*$^{-/-}$, 9 months: N = 8 and 13; *insl3*$^{+/+}$ and *insl3*$^{-/-}$, 12 months: N = 8 and 15; p < 0.05; **p < 0.01; ***p < 0.001). In **C**, yellow dashed lines indicate representative apoptotic germ cell cysts. Scale bars, 25 μm. **D, E** Quantitative analysis of spermatogenesis (**D**) and transcript levels of growth factors, steroidogenesis-related and Insl3 receptors (**E**) in 12 months-old *insl3*$^{+/+}$ and *insl3*$^{-/-}$ adult testis tissue. Data are mean ± SEM (*insl3*$^{+/+}$ and *insl3*$^{-/-}$: N = 6 and 10; *p < 0.05; **p < 0.01) and, in **E**, expressed as relative to the wild-type group (which is set at 1; dashed line). A$_{und}$, type A undifferentiated spermatogonia; A$_{diff}$, type A differentiating spermatogonia; B, type B spermatogonia; SPC, spermatocytes; SPT, spermatids; SPZ, spermatozoa; Others, including (i) empty spaces lined by Sertoli and/or germ cells that are not part of the lumen, (ii) Sertoli cell only areas, and (iii) apoptotic cells (see Supplementary Fig. 2 for further details); SC, Sertoli cell; LC, Leydig cell.

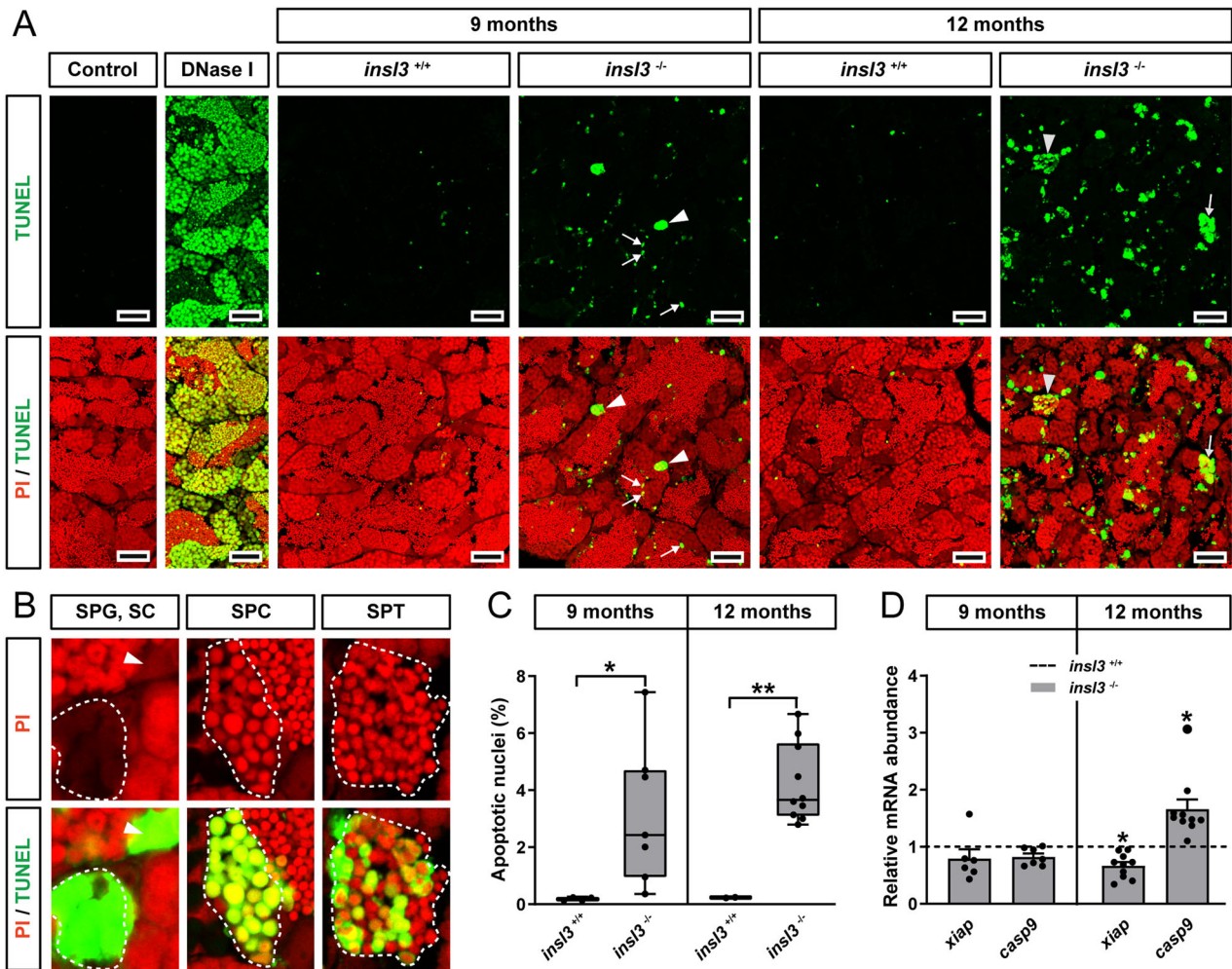

**Fig. 3 Confirmation of germ cell apoptosis/DNA damage in *insl3* knockouts by TUNEL analysis. A–C** Detection (**A**, **B**) and quantification (**C**) of germ cell apoptosis/DNA damage in wild-type (*insl3*$^{+/+}$) and *insl3* knockout (*insl3*$^{-/-}$) testes 9 and 12 months post-fertilization. In **A**, white arrowheads and arrows indicate representative TUNEL$^+$ spermatogonia and Sertoli cells, and gray arrowheads and arrows indicate representative TUNEL$^+$ spermatids and spermatocytes, respectively. In **B**, representative TUNEL$^+$ spermatogonia (SPG), spermatocyte (SPC) or spermatid (SPT) cysts are encircled with a white dashed line, and a TUNEL$^+$ Sertoli cell (SC) is indicated by a white arrowhead. TUNEL$^+$ cells/cysts are shown in green and propidium iodide (PI) counterstain is red. Scale bars, 25 μm. **D** Transcript levels of anti- (*xiap*) a pro-apoptotic (*casp9*) genes in *insl3*$^{+/+}$ and *insl3*$^{-/-}$ testis tissue 9 and 12 months post-fertilization. TUNEL quantification results are shown as mean ± SEM (*insl3*$^{+/+}$ and *insl3*$^{-/-}$, 9 months: N = 4 and 7; *insl3*$^{+/+}$ and *insl3*$^{-/-}$, 12 months: N = 3 and 10; *p < 0.05; **p < 0.01), and gene expression data as mean fold change ± SEM (*insl3*$^{+/+}$ and *insl3*$^{-/-}$, 9 months: N = 8 and 7; *insl3*$^{+/+}$ and *insl3*$^{-/-}$, 12 months: N = 6 and 10; *p < 0.05) and expressed relative to the wild-type group (which is set at 1).

**Insl3-induced changes in the testicular transcriptome**. Considering that Insl3 can exert direct effects on germ cells via Rxfp2a expressed by type A spermatogonia, and indirect effects via Rxfp2b expressed by Sertoli and myoid cells, and considering that loss of *insl3* resulted in a spermatogenesis phenotype associated with elevated germ cell apoptosis and changed expression of growth factor and steroidogenesis-related genes, we wanted to examine in a more comprehensive manner the biological activities of Insl3. To this end, we compared global gene expression of adult zebrafish testis tissue in response to Insl3 by RNAseq. Insl3 caused a significant modulation in the expression of 409 genes (Fig. 4A and Supplementary Data 1), with slightly more genes decreased (223 or ~55%) than increased (186 or ~45%), while the proportion of differentially expressed genes (DEGs) that reached a more than 2-fold change in expression was much higher for the Insl3-inhibited genes (194 or ~87%; Fig. 4A).

The majority of KEGG terms significantly enriched in Insl3-treated testes were down-regulated (Fig. 4B), including pathways related to steroid hormone biosynthesis, Ppar signaling, and retinol metabolism, as well as others involved in metabolic processes (e.g., glycolysis/gluconeogenesis, fatty acid, or pyruvate metabolism). Functional enrichment analysis revealed gene clusters characterized by a high number of overlapping genes including factors involved in sterol, lipid, and fatty acid metabolism (Fig. 4C). Among the candidates identified by functional analyses, transcript levels of all Ppar signaling genes were lowered by Insl3 (Fig. 4D). Similarly, ABC transporters, metabolic and steroid-related genes were preferentially down-regulated (Fig. 4D). On the contrary, a higher proportion of up-regulated genes was identified in the retinoid-related category (Fig. 4D). In addition, the gene set "Others" included factors involved in Wnt (*fzd8b*, *wisp3*, and *ccnd1*) as well as thyroid hormone signaling pathways (*tshba* and *thrb*; Fig. 4D), which were all up-regulated, except for *ccnd1*. Based on these data and on previous findings on the relevance of RA signaling for the differentiation of spermatogonia and meiosis in mammals[27] and

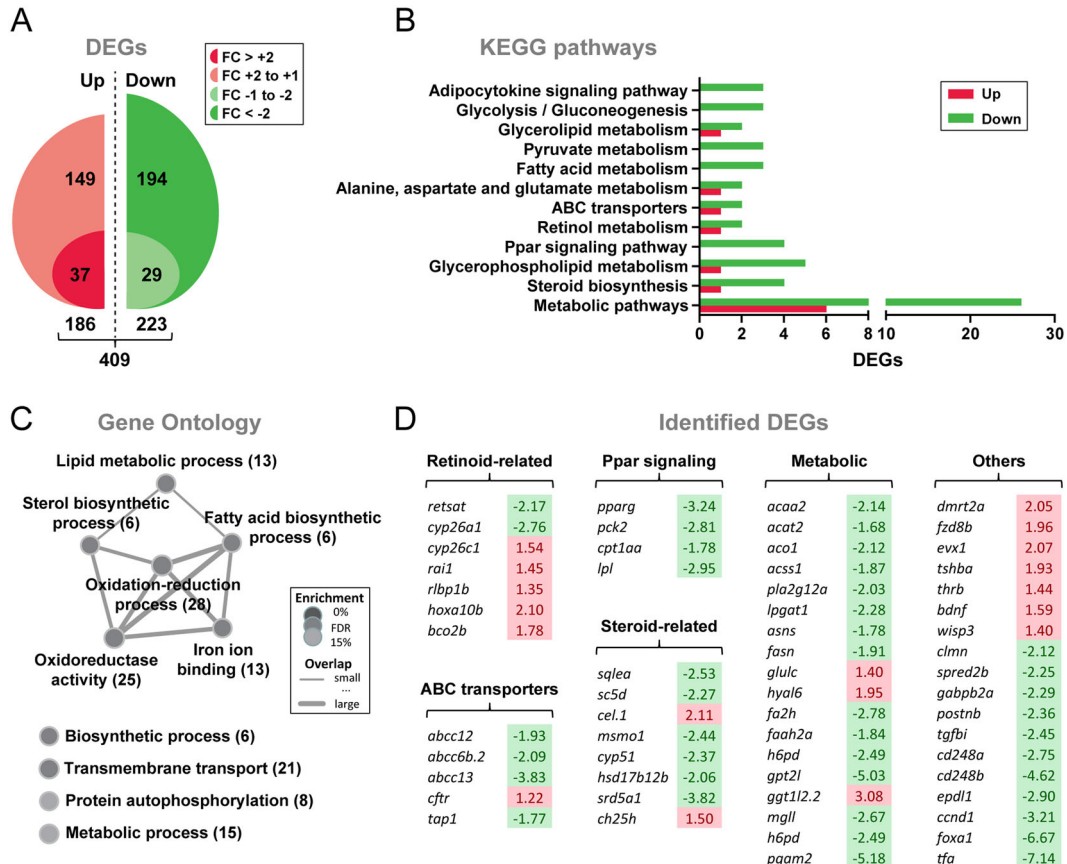

**Fig. 4 Gene expression profiling of testis tissue in response to Insl3. A** Total numbers of up-regulated and down-regulated genes (DEGs) identified by RNAseq ($N = 3$; $p < 0.05$). To generate testis samples for RNAseq, male zebrafish testes were incubated in the absence or presence of zebrafish Insl3 (100 ng/mL) for 2 days. FC fold change. **B**, **C** Insl3-regulated KEGG pathways (**B**) and Gene Ontology terms (**C**) in adult zebrafish testis tissue. KEGG pathways represented by at least three DEGs and ratio of regulated genes (down-regulated/up-regulated) higher than 2 were considered for the analysis. In **C**, number of identified genes is shown in brackets and enrichment significance (FDR, false discovery rate) is represented as a color gradient. **D** Selected DEGs identified by KEGG and GO analyses grouped by their function. Fold change values are shown with a red or green background indicating up-regulation or down-regulation, respectively.

zebrafish[25], we decided to investigate a possible link between Insl3 and RA signaling. Moreover, we were intrigued by the consistent modulation of Pparg expression.

**Insl3-induced effects on spermatogenesis are mediated by retinoic acid and Pparg signaling pathways.** To investigate the possible involvement of retinoid signaling in mediating Insl3 effects on zebrafish spermatogenesis, we incubated testicular explants with Insl3 (100 ng/mL) in the absence or presence of a RA production inhibitor (DEAB). Blocking testicular RA production elevated the BrdU index of undifferentiated spermatogonia (type $A_{und}$; Fig. 5A) and increased the proportion of area occupied by this cell type (Fig. 5B), suggesting that proliferation of type $A_{und}$ spermatogonia resulted in more type $A_{und}$ cells.

The proportion of differentiating spermatogonia (types $A_{diff}$ and B), on the other hand, was lowered when blocking RA synthesis in the presence of Insl3 (Fig. 5B). Since these cell types showed no change in proliferation activity (Fig. 5A), we understand their reduced proportion as reflecting the DEAB-induced reduction of RA-mediated pro-differentiation effects. Transcript levels of the RA-producing enzyme *aldh1a2* were not affected by Insl3, in contrast to the significantly reduced transcript levels of the RA-degrading enzyme *cyp26a1* (Fig. 5C), suggesting that Insl3 can increase testicular RA availability by decreasing its catabolism. An inhibitory effect of Insl3 on *cyp26a1* expression was also suggested by elevated levels of this transcript

in 12 months-old *insl3*$^{-/-}$ males in comparison with wild-type siblings (Fig. 5D). Analysis of RNAseq data that compared control, germ cell-depleted, and testis tissue recovering from this depletion[25], suggested that *aldh1a2* expression was enriched in somatic cells, since its transcript levels increased in germ cell-depleted testes and decreased again during the recovery of spermatogenesis (Fig. 5E). In contrast, no effect was observed for *cyp26a1* (Fig. 5E), indicating that somatic and germ cells express this enzyme, as suggested previously for zebrafish testis tissue based on in situ hybridization studies[28]. Taken together, changes in retinoid metabolism may mediate part of the Insl3 effects on spermatogenesis in zebrafish.

In view of the reduced levels of Ppar-related transcripts following Insl3 treatment (Fig. 4), we investigated possible interactions between Insl3 and Ppar signaling. First, we examined effects of the Pparg antagonist T0070907 in the primary testis tissue culture system. Distinct effects were recorded for undifferentiated versus differentiating spermatogonia: the Pparg antagonist halved the proportion of type $A_{und}$ but increased those of type $A_{diff}$ and B spermatogonia (Fig. 6A). Regarding BrdU incorporation, only the activity of type B spermatogonia increased in the presence of the Pparg antagonist (Fig. 6B). Similar to the effect of pharmacological Pparg inhibition, genetic loss of the *pparg* gene (allele 1737; *pparg*$^{-/-}$ $^{sa1737}$, see Supplementary Fig. 5 for more details on the two *pparg* mutant alleles) reduced the proportion occupied by type $A_{und}$ spermatogonia in homozygous

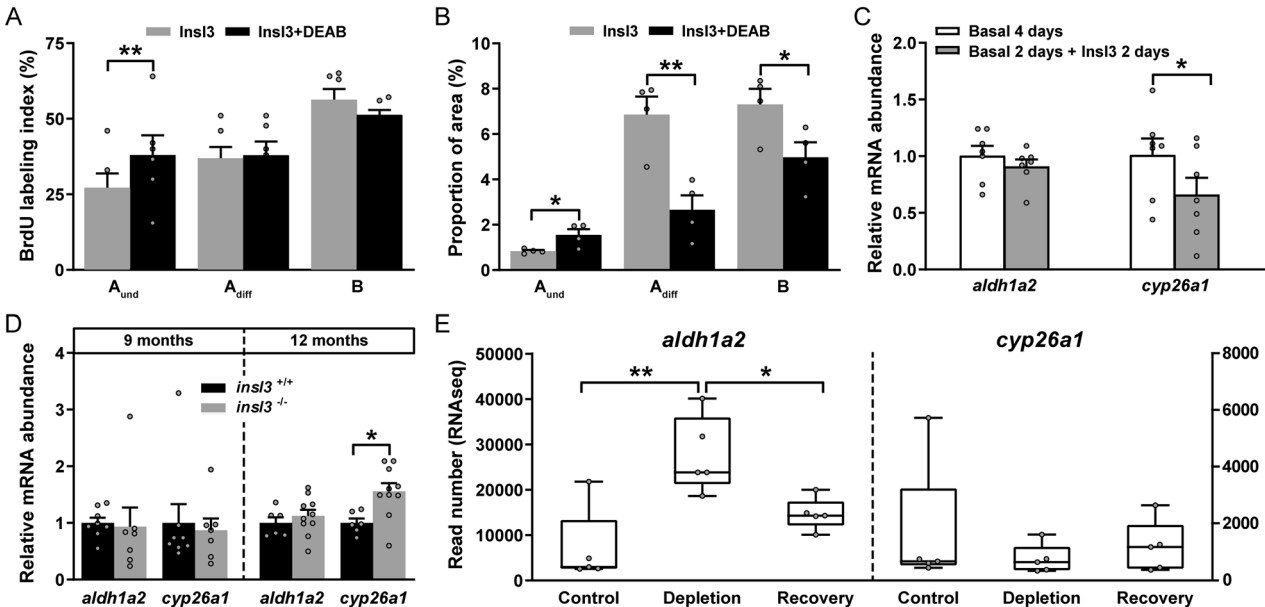

**Fig. 5 Involvement of retinoic acid (RA) signaling in Insl3-induced spermatogenesis. A**, **B** Evaluation of the proliferation activity (**A**) and of the proportions of spermatogonia (**B**) in zebrafish testes cultured for 4 days with 100 ng/mL Insl3, and in the absence or presence of the RA inhibitor DEAB (10 μM). **C** Ex vivo transcript levels of the RA producing (*aldh1a2*) and degrading enzymes (*cyp26a1*) in testis tissue incubated in the absence or presence of 100 ng/mL Insl3. **D** *aldh1a2* and *cyp26a1* expression levels in wild-type (*insl3*[+/+]) and *insl3* knockout (*insl3*[−/−]) testes 9 and 12 months post-fertilization. **E** Read numbers (RNAseq) of RA metabolic enzymes in control, germ cell-depleted (by exposure to the cytostatic agent busulfan[51]), and testes with recovering (from busulfan) spermatogenesis, as described by Crespo et al.[25] (NCBI GEO data set GSE116611). In **A**, **B** and **E**, data are shown as mean ± SEM (**A**: $N = 6$; **B**: $N = 4$; **E**: $N = 5$; $*p < 0.05$; $**p < 0.01$), and in **C**, **D** as mean fold change ± SEM (**C**: $N = 7$; **D**: *insl3*[+/+] and *insl3*[−/−] 9 months, $N = 8$ and 7; *insl3*[+/+] and *insl3*[−/−] 12 months, $N = 6$ and 10; $*p < 0.05$) and expressed relative to the control group (which is set at 1). A_und, type A undifferentiated spermatogonia; A_diff, type A differentiating spermatogonia; B, type B spermatogonia.

mutants (Fig. 6C). However, this effect was not observed in the mutant allele 1220 (*pparg*[−/− sa1220]; Fig. 6C). In testis tissue of 12 months-old *insl3*[−/−] mutants, *pparg* transcript levels were higher, while in 9 months-old mutants, lower levels were recorded compared to wild-type controls (Fig. 6D, left panel). Analyzing the RNAseq data set that compared control, germ cell-depleted, and recovering testes[25], showed that *pparg* expression was enriched in somatic cells, since its transcript levels increased in germ cell-depleted testes and decreased again during the recovery of spermatogenesis (Fig. 6D, right panel). Taken together, these results suggest that Insl3 supports spermatogenesis in the adult zebrafish testis by reducing Pparg signaling in somatic cells.

Despite functional enrichments for steroid-related genes in our RNAseq data (Fig. 4), further analyses did not support a direct effect of Insl3 on zebrafish testicular steroidogenesis. Neither androgen (i.e., 11-ketotestosterone [11-KT]) production nor transcript levels of selected genes involved in steroidogenesis responded to zebrafish Insl3 in primary tissue culture (Supplementary Fig. 6A, B). Furthermore, the Insl3-triggered stimulation of the proliferation of spermatogonia was not modulated by preventing the production of biologically active steroids by trilostane (TRIL; Supplementary Fig. 6C), demonstrating that steroid signaling does not mediate acute Insl3 effects on germ cell proliferation.

## Discussion

After having found in previous work that Insl3 promoted the differentiating division of spermatogonia, we investigated how Insl3 stimulated germ cell differentiation. We found that Insl3 (i) used two receptor paralogues expressed by Sertoli and myoid cells (Rxfp2b) and type A spermatogonia (Rxfp2a), respectively; (ii) reduced germ cell apoptosis also in zebrafish; (iii) stimulated the proliferation activity of type A_und spermatogonia using a so far

unknown mechanism, however, not involving RA or Pparg; and (iv) increased the transition of A_und spermatogonia to A_diff and B spermatogonia via enhancing RA and reducing Pparg signaling.

Of the four candidate Insl3 receptors, previously identified in zebrafish testis tissue[24], two responded well to Insl3, with Rxfp2b being ~15-fold more sensitive than Rxfp2a. Rxfp2b also showed a higher level of constitutive activity, and was expressed by somatic cells in close contact with germ cells, probably Sertoli and myoid cells, while Rxfp2a was expressed by type A spermatogonia. Also in zebrafish, Leydig cells are the cellular source of Insl3[23]. Considering the proximity of Leydig and Sertoli and myoid cells, the high sensitivity to Insl3 and the high basal activity of Rxfp2b, the latter is likely to signal at least somewhat most of the time. Type A spermatogonia, on the other hand, are shielded from Insl3 to some extent by cytoplasmic extensions of Sertoli cells, and the Rxfp2a receptor variant these germ cells express requires higher Insl3 concentrations for activation. Therefore, it seems possible that the higher Insl3 concentration required to activate Rxfp2a expressed by type A spermatogonia, is only achieved following an Fsh stimulus of Insl3 production[21]. Studies in mammals also reported germ cell expression of *Rxfp2* mRNA, but then restricted to spermatids in rat[15], mice[7], and boar[29], while in the latter, also expression in spermatocytes was reported.

In young adult males at 3 and 6 months of age, the knockout of *insl3* went unnoticed. However, at and beyond 9 months of age, we observed increased apoptotic activity among germ cells, reduced GSI and reduced proportions of the more advanced germ cell generations type B spermatogonia, spermatocytes, and spermatids. Our morphological data suggest that scattered single germ cells or small groups of germ cells were lost to apoptosis initially, which may have resulted in a number of small and then, perhaps by confluence, larger empty spaces in the germinal epithelium. These spaces were first bounded by the cytoplasmic

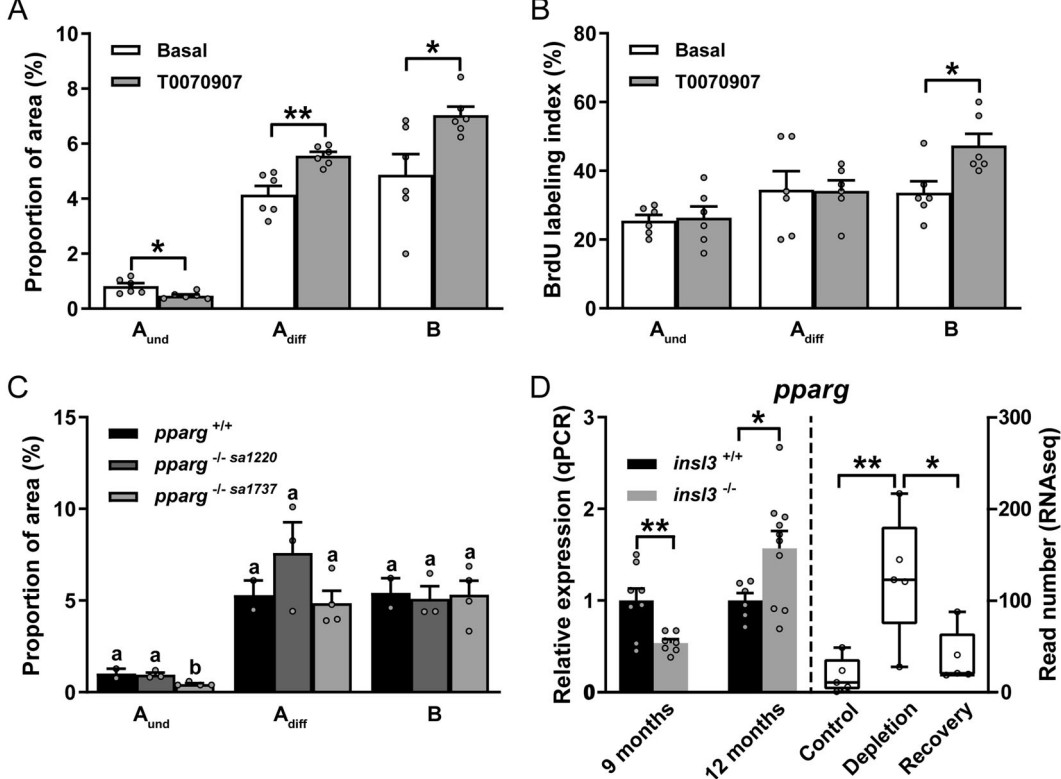

**Fig. 6 Pparg involvement in Insl3-induced spermatogenesis. A**, **B** Evaluation of the proportions (**A**) and of the proliferation activity of spermatogonia (**B**) in zebrafish testes cultured for 4 days in the absence or presence of the Pparg antagonist T0070907 (10 μM). **C** Area occupied by different types of spermatogonia in wild-type (*pparg*[+/+]) and *pparg* knockout (*pparg*[−/−]) adult testes. Two different *pparg*[-/-] mutants (alleles 1220 and 1737) were evaluated. **D** In vivo *pparg* expression levels in wild-type (*insl3*[+/+]) and *insl3* knockout (*insl3*[-/-]) testes 9 and 12 months post-fertilization (left panel), and in control, germ cell-depleted (by exposure to the cytostatic agent busulfan[51]), and testes with recovering (from busulfan) spermatogenesis, as described by Crespo et al.[25] (NCBI GEO data set GSE116611) (right panel). In **A**–**C** and right panel in **D**, data are shown as mean ± SEM (**A**, **B**: $N = 6$; **C**: *pparg*[+/+], *pparg*[−/− sa1220] and *pparg*[−/− sa1737], $N = 2$, 3 and 4; **D**: $N = 5$; *$p < 0.05$; **$p < 0.01$), and in the left panel in **D** as mean fold change ± SEM (*insl3*[+/+] and *insl3*[−/−], 9 months: $N = 8$ and 7; *insl3*[+/+] and *insl3*[−/−], 12 months: $N = 6$ and 10; *$p < 0.05$; **$p < 0.01$) and expressed relative to the control group (which is set at 1). In **C**, different letters indicate significant differences between groups (*$p < 0.05$). A$_{und}$, type A undifferentiated spermatogonia; A$_{diff}$, type A differentiating spermatogonia; B, type B spermatogonia.

extensions of Sertoli cells, but these spaces may be lost eventually, perhaps by fusing with the tubular lumen, while the remaining Sertoli cells may give rise to the Sertoli cell only groups. This observation would also indicate that, in the long run, spermatogonial stem cells of *insl3*[−/−] mutants are unable to replenish the lost germ cells, or that mutant Sertoli cells are no longer able to efficiently produce new spermatogenic cysts.

In boar[13] and rat[15], but not in mice[7], INSL3 was considered an anti-apoptotic factor for germ cells. This is reminiscent of the situation in zebrafish, so that a conserved function of Insl3 seems to be to reduce germ cell apoptosis via receptors expressed by germ cells. However, we found no evidence for *rfxp2a/b* expression in zebrafish spermatocytes and spermatids. There are number of possibilities to understand this apparent mismatch. Transgene expression may not be fully representative of the native promotor activity, or transgene expression at later stages of spermatogenesis is weak and not easily detectable. Unfortunately, in situ hybridization to locate *rfxp2a* mRNA on testis sections was not successful in our hands, possibly related to low levels of this transcript; median read numbers were not exceeding 100 in our RNAseq study (Fig. 1B). Future work using potentially more sensitive approaches[30,31] is warranted in this regard. While not having data on the exact cellular localization of the transcripts, examining their levels after germ cell depletion/recovery demonstrated that *rfxp2a* mRNA resided in the germ cell compartment, whereas *rfxp2b* mRNA is expressed in the testicular

somatic compartment. Finally, reporter protein expression may not represent reliably a possibility often encountered in germ cells, namely the storage of precociously expressed mRNA for later use[32]. Sertoli cell-associated *rxfp2b* expression, on the other hand, is unlikely to be involved in the observed germ cell apoptosis, since in the cystic type of spermatogenesis in fish, Sertoli cell-mediated apoptosis would be expected to affect all germ cells in a given germ cell clone, which was not observed in our study.

Increased apoptosis probably contributed to the reduced volume fractions measured for spermatocytes and spermatids at 12 months of age (Fig. 2D), but apoptosis did not affect type B spermatogonia, so that their reduced volume fraction in *insl3*[−/−] mutants must have a different background. In this regard, it seems relevant that Insl3 promoted the differentiating division of type A$_{und}$ to type A$_{diff}$ spermatogonia in primary testis tissue cultures of adult zebrafish[20,23]. Removing this stimulatory effect in vivo may eventually result in a reduced production of type B spermatogonia. In 9 months-old *insl3*[−/−] males, transcript levels of the key enzyme for androgen production were 3-fold upregulated (Supplementary Fig. 4). The androgen 11-KT stimulated spermatogenesis in zebrafish[25,33] and loss of the androgen receptor gene resulted in hypoplastic testes and disturbed spermatogenesis[34,35]. Therefore, the potentially elevated androgen production may have counterbalanced in part the absence of Insl3. Spermatogenesis further deteriorated in 12 compared to 9 months-old mutants. At that time, *cyp17a1* transcript had fallen

behind the controls (Fig. 2E), suggesting that it will be interesting in future studies to compare androgen plasma levels and testicular androgen levels in wild-type and $insl3^{-/-}$ mutant males of 9 and 12 months of age. The decrease in $cyp17a1$ transcript levels at 12 months of age was accompanied by reduced $igf3$ mRNA levels, a growth factor stimulating the differentiation of spermatogonia and their entry into meiosis[36,37]. Moreover, key enzymes controlling RA levels (in turn promoting germ cell differentiation; see below), were expressed at similar levels in 9 months-old wild-type and mutant testes but shifted in 12 months-old mutants to facilitate RA breakdown (Fig. 5D). Finally, Pparg (restricting differentiation of $A_{und}$ and reducing the production of type B spermatogonia; see below), were down- and up-regulated, respectively, in 9 and 12 months-old mutants (Fig. 6D). Jointly, these observations suggest that up until ~9 months of age, mutant testes compensated the loss of the pro-differentiation factor Insl3 by reducing signaling that restricts differentiation of $A_{und}$ spermatogonia (Pparg), and by sustaining (Igf3, RA) or increasing (androgen) pro-differentiation signals. The concept of a long-term, compensatory reaction, instead of a direct Insl3-regulated short-term response, is supported by the observation that in short-term testis tissue culture studies[20,23], Insl3 had no effect on transcript levels of growth factor or steroidogenesis-related genes. The latter was confirmed again in the present study, now also showing that biochemical blocking of androgen production did not modulate the action of Insl3 on the proliferation activity of type A spermatogonia (Supplementary Fig. 6C). While the biological activity of Insl3 does not seem to depend on androgen production or action, blocking androgen production genetically clearly reduced Insl3 production, so androgens may be up-stream of Insl3[38]. However, developmental effects of androgen insufficiency on Leydig cell number and/or maturation are possible and may secondarily reduce Insl3 production. Taken together, sex differentiation, puberty, and spermatogenesis in young adults proceeded phenotypically normally in $insl3^{-/-}$ males, potentially involving the activation of compensatory mechanisms. However, in older adults ≥9 months of age, the compensatory mode became exhausted, leading to a deterioration of spermatogenesis.

The $gsdf$ gene is not required for male fertility[39], is expressed by Sertoli cells contacting all stages of germ cell development[40], and does not respond to Fsh or Lh in zebrafish[21]. We therefore use it here as an indicator of Sertoli cell number. In this regard, the progressively lower $cx43$ transcript levels in $insl3$ mutants, despite stable $gsdf$ transcript levels, indicate that Sertoli cell gap junctions, but not Sertoli cell number, may have been compromised. In the adult mammalian testis, the gap junction protein CX43 is relevant for the communication among neighboring Sertoli cells, and is required specifically also for the integrity of the tight junctions that are established among Sertoli cells during puberty and after Sertoli cells stopped proliferating and differentiated terminally[26]. In zebrafish, however, these junctions are not formed throughout the testis between all Sertoli cells during puberty but are established only among Sertoli cells of those spermatogenic cysts, in which the germ cells approach the end of meiosis[41]. We speculate that a reduced availability of Cx43 disturbed the communication and/or establishment of tight junctions between Sertoli cells enveloping late spermatocytes/spermatids, which may have reduced Sertoli cell functionality, and thereby contributed to the increased apoptotic loss of meiotic and postmeiotic germ cells. Sertoli cell-specific loss of $Cx43$ in mice also was associated with a failure of spermatogonia to differentiate[42], so that reduced $cx43$ transcript levels may contribute to a reduced production of differentiating spermatogonia in $insl3^{-/-}$ zebrafish.

As discussed above, increased transcript levels encoding the RA-catabolizing enzyme Cyp26a1 suggested a reduced availability

of RA in 12 months-old mutant testis tissue. RA as ligand for its Raraa receptor is relevant for supporting spermatogenesis in zebrafish, which includes the restriction of apoptosis among spermatocytes and in particular spermatids[25]. Analysis of our RNAseq data confirmed Insl3 regulation of retinoid-related transcripts. Direct experimental evidence for the interaction of Insl3 and RA-mediated signaling is provided by the Insl3-induced decrease in $cyp26a1$ transcript levels in primary testis tissue culture (Fig. 5C), fitting well to increased $cyp26a1$ transcript levels in 12 months-old $insl3^{-/-}$ mutants discussed above. We therefore propose that part of the biological activity of Insl3 is mediated via RA signaling through Raraa in zebrafish testis tissue.

What aspect of Insl3 activity may be related to RA signaling, next to the Raraa-mediated effects on spermatid apoptosis discussed above? When only blocking RA production, neither the BrdU index nor the proportion of area changed for type $A_{und}$[25]. Only adding Insl3 to testis tissue, increased the BrdU index of type $A_{und}$, while their proportion of area decreased[20]. Here, we found that blocking RA production in the presence of Insl3 further increased the BrdU index, but now also increased the proportion of area for type $A_{und}$ (Fig. 5A, B). Jointly, these observations indicate that Insl3 increased the BrdU index of $A_{und}$, while also facilitating RA production, thereby promoting differentiation of the newly formed germ cells[25]. The latter also explains the combination of partial depletion of $A_{und}$ and accumulation of $A_{diff}$ that did not show a change in their BrdU index[20]. Vice versa, when blocking RA production in the presence of Insl3, the proportion of $A_{diff}$ is decreased, probably due to a shortage of RA, so that $A_{und}$ produced under the influence of Insl3 remain undifferentiated and accumulate (Fig. 5B). Taken together, our results suggest that the previously described effect of Insl3 to promote the differentiating division of $A_{und}$[20,23] is composed of at least two separate processes: (i) the Insl3-triggered stimulation of $A_{und}$ cell cycling that is undisturbed by DEAB/RA; and (ii) the DEAB/RA-sensitive guidance of the newly formed cells into differentiation, the latter potentially also supported by Sertoli cell to spermatogonia communication involving Cx43-containing junctions. Missing this Insl3-mediated stimulation of $A_{und}$ cell cycling in mutants may also explain the appearance of the Sertoli cell only patches in testis tissue of older mutants.

In addition to retinoid signaling, Ppar signaling was retrieved from the RNAseq analysis, with Pparg as the leading gene, a member of the nuclear receptor family. PPARG has a broad ligand binding spectrum, including unsaturated fatty acids, eicosanoids, and the prostaglandin $PGJ_2$[43]. In the human testis, PPARG protein is found in Sertoli cells and in germ cells (spermatocytes and spermatozoa)[44]. However, information available in the human Protein Atlas database[45] also indicates interstitial/extratubular PPARG/$Pparg$ expression. Our RNAseq data in zebrafish suggests somatic but not germ cell $pparg$ expression (Fig. 6D). Unfortunately, in situ hybridization trials were not successful, possibly related to the low read numbers (~10; controls in Fig. 6D) found for this transcript. To develop the biological activity, PPARG forms heterodimers with a retinoic X receptor. All six $rxr$ paralogues in the zebrafish genome are expressed in the testis, and the expression pattern of two of these paralogues ($rxrab$ and $rxrgb$) suggests preferential somatic expression (Supplementary Fig. 7), while the four other paralogues were expressed in both, somatic and germ cells[25]. It therefore seems likely that Pparg can interact with a retinoic X receptor in zebrafish Sertoli cells.

PPARG is known for regulating adipogenesis, energy balance, and lipid biosynthesis[43]. There are no adipocytes in the spermatogenic tubules or the interstitial tissue, but PPARG also regulates adipocyte differentiation from mesenchymal stem cells[46]. Other stem cell systems were reported to be sensitive to

PPARG signaling as well[47–49], so we speculate that Pparg signaling may affect stem cell populations in the zebrafish testis. We have postulated previously[50] that a somatic stem cell population in the fish testis gives rise to Sertoli cells, which may be a target of Pparg signaling in addition to targeting processes in differentiated Sertoli cells. The established testicular stem cell type, spermatogonial stem cells, belongs to the population of type $A_{und}$ spermatogonia[51] and would have to be affected indirectly via Pparg-mediated changes in Sertoli cell activity. In this regard, it is interesting to note that in rodent Sertoli cells PPARG regulates lipid storage and lactate production[52]. These metabolic activities are important for meeting the energy demand of germ cells under the relative hypoxic conditions in the germinal epithelium, and reduce reactive oxygen species (ROS) production associated with oxidative energy production, which seems particular relevant considering the ROS-sensitivity of stem cells[53].

We found that pharmacological (Pparg inhibitor T0070907) as well as genetic ($pparg^{-/- \ sa1737}$) interference with Pparg activity, removed a protection of type $A_{und}$ spermatogonia against pro-differentiation effects, resulting in a partial loss of $A_{und}$ (Fig. 6B, C). The stimulatory effect on the BrdU-index and proportion of area of type B spermatogonia, on the other hand, was visible in the short-term pharmacological experiments, but not in the long-term genetic model. It is possible that this effect was lost in context with the compensatory responses of the mutant testis tissue. Taken together, our observations suggest that Pparg modulates spermatogenesis in adult zebrafish in two ways: (i) Pparg reduces the $A_{und}$ to $A_{diff}$ transition, apparently without changing their proliferation activity, thus reducing the production of $A_{diff}$; (ii) under short-term conditions, Pparg can reduce the proliferation activity and hence number of type B spermatogonia. It appears that Pparg can tilt the balance of germ cell development in favor of keeping type A spermatogonia in an undifferentiated state while reducing the number of more differentiated spermatogonia. The second $pparg$ mutant allele ($pparg^{-/- \ sa1220}$) did not show a phenotype. We speculate that the absence of a phenotype of the $pparg^{-/- \ sa1220}$ allele (see Supplementary Fig. 5C for an alignment with the human protein; functional information on zebrafish $pparg$ mutant alleles is not available) may reflect the complete absence of the domains required for DNA binding, combined with its potential replacement by other Ppar family members. The $pparg^{-/- \ sa1220}$ allele, on the other hand, retained the first zinc finger and the P-box but missed the D-box required for dimerization, so that it may show traits of a dominant negative protein reducing access of Ppar proteins to their binding sites on the DNA. However, these speculations require further studies.

In summary, increased apoptotic loss of germ cells after loss of $insl3$ suggests that an anti-apoptotic effect is among the evolutionary conserved Insl3 functions, while we found no evidence for an acute effect on testicular steroidogenesis in zebrafish. RNAseq data and follow-up studies showed that RA and Pparg signaling mediated Insl3 effects, resulting in the increased production of differentiating spermatogonia in response to Fsh-stimulated Insl3 production (schematically summarized in Fig. 7). However, Insl3 effects are not drastic, and testicular defects in $insl3$ mutants are not noticeable initially. Overall, previous and present results show that Fsh uses different, locally produced signaling molecules (growth factors, including Insl3, but also low molecular weight molecules like RA[25], sex steroids[34,35] and prostaglandins[54]) to implement specific regulatory effects on spermatogenesis. Since several of these Fsh-regulated pathways operate in parallel in zebrafish, the impact of an individual pathway usually is not overwhelming, allowing follow-up research to examine compensatory mechanisms. This seems different in a number of cases in mammals considering that for example androgen receptor and

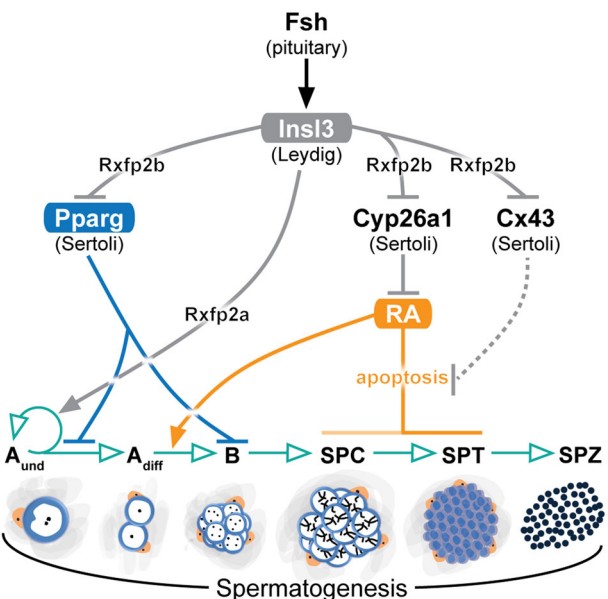

**Fig. 7 Schematic illustration showing the endocrine and paracrine regulation of zebrafish Insl3 and the stages of spermatogonial development affected.** Described effects are indicated by black (Fsh), gray (Insl3), blue (Pparg), and orange (RA) arrows, while germ cell development or germ cell-mediated effects are indicated in green. Gray dashed line denotes no experimental evidence reported. Fsh, follicle-stimulating hormone; Insl3, insulin-like 3; Pparg, peroxisome proliferator-activated receptor gamma; RA, retinoic acid; Cx43, connexin 43; Rxfp2a, relaxin family peptide receptor 2a; Rxfp2b, relaxin family peptide receptor 2b; $A_{und}$, type A undifferentiated spermatogonia; $A_{diff}$, type A differentiating spermatogonia; B, type B spermatogonia; SPC, spermatocytes; SPT, spermatids; SPZ, spermatozoa; Leydig, Leydig cell; Sertoli, Sertoli cell.

RA receptor gamma each individually are indispensable for spermatogenesis[55,56]. This creates "bottlenecks" that do not exist in fish, although the same signaling systems are relevant in spermatogenesis throughout vertebrates.

## Methods

**Fish maintenance.** Zebrafish were bred and raised in the aquarium facility of the Faculty of Science at Utrecht University (The Netherlands). Sexually mature males between 3 and 12 months of age were used for the present experiments. Handling and experimentation were consistent with the Dutch national regulations. The Life Science Faculties Committee for Animal Care and Use in Utrecht approved the experimental protocols.

**Identification of Insl3 responsive receptors.** Zebrafish testis expressed four candidate Insl3 receptor genes: $rxfp1$, $rxfp2a$, $rxfp2b$, and $rxfp2-like$[24]. The coding regions of each of these receptors was cloned into pcDNA3.1/V5-His vector and assayed to mediate zebrafish Insl3-stimulated cAMP-induced reporter-gene activity according to Chen et al.[57], with minor modifications as described previously[58]. Briefly, human embryonic kidney (HEK-T) 293 cells were maintained under 5% $CO_2$ and at 37 °C in culture medium (Dulbecco's modified Eagle's medium [DMEM]) containing 2 mM glutamine, 10% fetal bovine serum and 1× antibiotic/antimycotic solution (all from Invitrogen). Transient transfections were performed in 10 cm dishes, containing approximately $3.5 \times 10^6$ cells with 1 μg receptor expression vector construct in combination with 10 μg pCRE/β-gal plasmid, 66 μg polyethylenimine (PEI; Polysciences Inc.) and 150 nM NaCl in D-PBS and diluted in culture medium following overnight incubation. The pCRE/β-gal plasmid consists of a β-galactosidase gene under the control of a human vasoactive intestinal peptide promoter containing five cAMP-response elements[57]. Empty pcDNA3.1/V5-His vector was used for mock transfections. The next day, the cells were stimulated with increasing concentrations of recombinant Insl3[23] in HEPES-modified DMEM containing 0.1% BSA and 0.1 mM IBMX (all from Sigma). Ligand-induced changes in β-galactosidase activity (conversion of o-nitrophenyl-β-D-galactopyranoside into o-nitrophenol) were measured at 405 nm in a Bio-Rad 96-well microplate reader, and related to the forskolin (10 μM)-induced changes in each 96-well plate. Therefore, the results are expressed as arbitrary units (AU, Fig. 1A), related to the forskolin-induced cAMP-mediated reporter gene activation.

All experiments were repeated at least three times using cells from independent transfections, each performed in triplicate. Ligand concentrations inducing a half-maximal stimulation ($EC_{50}$) were calculated using GraphPad Prism (GraphPad Software, Inc.).

**Generation of *rxfp2a* and *rxfp2b* transgenic lines**. Rxfp2a and Rxfp2b were consistently activated by Insl3 with $EC_{50}$ values of 96.2 ng/mL and 6.5 ng/mL, respectively (Fig. 1A), while Rxfp1 and Rxfp2-like needed higher doses and did not reach the same level of stimulation. To study which cell types express these two receptors in testis tissue, we generated transgenic zebrafish lines expressing fluorescent proteins (FP) (*Tg(rxfp2a:EGFP)* and *Tg(rxfp2b:mCherry)*) under the control of approximately 3 kb of their promoter sequences in destination vectors pDestTol2CG2 (containing *cmlc2:EGFP*[59]) or pDestTol2CmC2 (containing *cmlc2:mCherry*; i.e., pDestTol2CG2, in which the EGFP sequence was replaced by the mCherry sequence), using Gateway technology[59]. To this end, zebrafish genomic DNA was amplified using primers 4703 and 4704, and 4705 and 4706 (see Supplementary Table 1), to specifically amplify *rxfp2a* (3193 bp) and *rxfp2b* (3087 bp) sequences, respectively, preceding the ATG start codons of these genes. These sequences were cloned into p5E-Fse-Asc vector[59]. Subsequent LR clonase reactions generated constructs for Tol2 transposase-mediated transgenesis, according to Ishibashi et al.[60]. Briefly, freshly fertilized, one-stage zebrafish embryos (AB strain) were injected with 1–2 nL of 50 ng/μL plasmid DNA and 100 ng/μL transposase mRNA. Three days after injection, embryos were selected based on the green or red fluorescence of their hearts, driven by the *cmlc2* sequence. Positive F0 fish were grown to adulthood and crossed with wild-type fish to establish transgenic lines.

**Generation of *insl3* and source for *pparg* knockout mutants**. CRISPR/Cas9 targets for the zebrafish *insl3* gene were selected using ZiFiT Targeter software. Guide RNA and Cas9 synthesis were performed according to Jao et al.[61]. For gRNA synthesis, the template DNA was linearized by *Bsm* BI digestion followed by in vitro transcription with MEGAscript T7 Kit (Ambion). The transcribed RNA was purified using a QIAprep Spin Miniprep kit (Qiagen) following the manufacturer's instructions. Plasmids sequences were verified by PCR amplification and Sanger sequencing. Cas9 mRNA was synthesized using a mMESSAGE mMA-CHINE Kit (Ambion) and linearized plasmid DNA as template. The resultant mRNA was purified before resuspension in nuclease-free water and quantified using a NanoDrop (Thermo Scientific). Freshly fertilized zebrafish embryos (Tüpfel long fin) were at the one-cell stage were co-injected with gRNA and Cas9 mRNAs (25 and 300 ng/μL, respectively). Injected embryos were maintained in an 28 °C incubator and transferred to aquarium tanks at 72 h of postinjection.

Mutant lines for *pparg* (sa1220 and sa1737) were obtained from the Zebrafish International Resource Center (ZIRC). Both mutant *pparg* alleles have an A>T nonsense mutation in exon 3 leading to a premature stop codon at either amino acid 141 (*pparg*$^{-/-\ sa1220}$) or amino acid 176 (*pparg*$^{-/-\ sa1737}$). Genotyping was performed using PCR-based KASP (Kompetitive Allele-Specific PCR) technology (LGC Genomics) performed on a Bio-Rad CFX96 machine according to the manufacturer's instructions (*pparg*$^{-/-\ sa1220}$ KASP assay ID: 554-1129.1; *pparg*$^{-/-\ sa1737}$ KASP assay ID: 554-1682.1). In the case of both alleles, the work presented here was carried out using the F4 generation after three outcrosses against wild-type AB fish, so that background mutations are unlikely to play an important role for our observations.

**Primary testis tissue cultures**. Using a previously established ex vivo tissue culture system[33], adult zebrafish testis tissue was incubated for 4 days with Insl3 alone (100 ng/mL)[20] or in the presence of also 10 μM *N,N*-diethylamino-benzaldehyde (DEAB, Sigma-Aldrich; a compound blocking RA production[62,63]). This served to investigate if RA signaling is involved in mediating Insl3 effects on spermatogonia proliferation and differentiation. Additional tissue culture experiments were carried out to examine the effect of Pparg signaling on zebrafish spermatogenesis, incubating testis tissue for 4 days in the absence or presence of 10 μM 2-chloro-5-nitro-N-4-pyridinyl-benzamide (T0070907, Sigma-Aldrich; a PPARg antagonist[64,65]), in order to evaluate its effects on spermatogonia proliferation and differentiation. Moreover, testis tissue was incubated for 18 h or for 4 days in the absence or presence of 100 ng/mL Insl3 to study its potential effects on androgen production and steroid-related gene expression, respectively. To exclude the potential effects of steroid hormones on the Insl3-stimulatory action on spermatogonia proliferation, additional incubations were carried out for 4 days in the presence of 25 μg/mL trilostane (TRIL, Sigma-Aldrich), which prevents the production of biologically active steroids. In all experimental conditions described above, testis tissue was incubated at 26 °C and experiments were repeated 2–3 times (including 5–12 individuals each).

Primary testis tissue cultures were also used to examine Insl3-induced changes in testicular gene expression by RNA sequencing (see below for details on RNA sequencing). Previous work showed that Insl3 changed testicular transcript levels (e.g., *aldh1a2* or *cyp26a1*) after incubating testis tissue for 4–7 days[20,23]. However, we considered 4–7 days as a period too long to identify direct, or at least not very far downstream, Insl3 target genes. We hypothesized that this relatively long period was necessary due to high levels of basal *insl3* gene expression in zebrafish Leydig cell (~15,600 reads; RNAseq data set GSE116611). In analogy to the spontaneous

decrease of sex steroid production to less than 10% of starting levels within two days after starting primary tissue culture[33], we speculated that *insl3* gene expression would wane in a similar manner. Accordingly, we first incubated testis tissue under basal conditions for 2 days to allow for the assumed decrease of endogenous Insl3 production, then added exogenous, zebrafish Insl3 peptide (100 ng/mL) and continued the incubation for another 2 days, before collecting the tissue for analysis of gene expression. The period of 2 days was chosen based on previous work on Fsh-induced changes in testicular gene expression, which also made use of a 2 days long incubation period[20]. Pilot studies showed under these conditions, we indeed recorded a significant decrease of *cyp26a1* transcript levels after only 2 days of exposure to Insl3.

**Transcriptomic analysis of testis tissue using RNA sequencing**. Total RNA from three control and three Insl3-treated testes was isolated using the miRNeasy Mini Kit (Qiagen). RNA integrity was checked with an Agilent Bio-analyzer 2100 total RNA Nano series II chip (Agilent). All six samples showed an RNA integrity number > 8 and were used for library preparation. Illumina RNAseq libraries were prepared from 2 μg total RNA using the Illumina TruSeq RNA Sample Prep Kit v2 (Illumina, Inc.) according to the manufacturer's instructions. The resulting RNAseq libraries were sequenced on an Illumina HiSeq2500 sequencer (Illumina, Inc.) as paired-end 150 nucleotide reads. Image analysis and base calling were done by the Illumina pipeline. Quality control of the obtained reads was performed using CASAVA software (v1.8; Illumina, Inc.). The sequencing yield ranged between ~41 and ~52 million reads per sample and mapping efficiency for uniquely mapped reads was between 69.2 and 71.9% (see Supplementary Data 1). RNAseq derived reads were aligned to the zebrafish genome (GRCz10) using TopHat[66] (v2.0.12). The resulting read counts were extracted using the Python package HTSeq[67] (v0.6.1). Data analysis was performed with the R/Bioconductor package DESeq[68] (v.1.18.0; $p < 0.05$). The raw RNAseq data of the six samples sequenced (three biological replicates per condition) have been deposited in the NCBI GEO database with accession number (GSE152038).

Functional enrichment analyses were carried out using a plugin[69] for the Cytoscape network environment[70]. The Enrichment Map plugin calculates over-representation of genes involved in closely related Gene Ontology (GO) categories[71], resulting in a network composed of gene sets grouped according to their function. DAVID Bioinformatics Resources 6.7[72] was used to retrieve GO terms from the list of differentially expressed genes (DEGs) and exported as the input for each functional enrichment analysis. Regulated KEGG pathways were determined using the KEGG Mapper tool[73]. KEGG pathways represented by at least three DEGs and by the ratios of regulated genes (up-/down-, and vice versa) higher than two were considered for the analysis.

**Testis tissue sample preparation and analysis: morphology, candidate gene expression, and androgen release**. The proliferation activity of type A and type B spermatogonia was quantified by examining the incorporation of the S-phase marker bromodeoxyuridine (BrdU; 50 μg/mL, Sigma-Aldrich), which was added to the medium during the last 6 h of the culture period. After incubation, testis tissue was fixed at room temperature for 1 h in freshly prepared methacarn (60% [v/v] methanol, 30% chloroform and 10% acetic acid glacial; Merck Millipore). Testis tissue was dehydrated, embedded in Technovit 7100 (Heraeus Kulzer), and sectioned at a thickness of 4 μm, according to conventional techniques. The mitotic index was determined by examining at least 100 germ cells (A$_{und}$) or spermatogenic cysts (A$_{diff}$ or B spermatogonia), differentiating between BrdU-labeled and unlabeled cells/cysts as previously described[25]. To quantify the proportion of area occupied by different germ cell types, testis tissue was fixed in 4% glutaraldehyde (4 °C, overnight), dehydrated, embedded in plastic, sectioned at 4 μm thickness and finally stained with toluidine blue. 10–15 randomly chosen, non-overlapping fields were photographed at ×400 magnification and the images were analyzed quantitatively using ImageJ software. With a specific plugin, a 540-point grid served to quantify the proportion of area occupied by the various germ cell types, based on the number of points counted over those germ cell types. The germ cells/cysts were identified according to previously published morphological criteria[41].

Additional sets of ex vivo experiments were carried out to investigate candidate gene expression in response to the same conditions used for generating tissue samples for the above-mentioned morphological analyses. To this end, total RNA was isolated from testis tissue at the end of the incubation period, using the RNAqueous Kit (Ambion) following the manufacturer's instructions. Relative mRNA levels of candidate genes were quantified by real-time, quantitative PCR (qPCR; see Supplementary Table 2 for detailed primer information) as previously described[21]. The geometric mean of *eef1a1l1*, *rpl13a*, and *ubc* was used as endogenous (housekeeping) control due to their constant expression under all conditions investigated. The relative mRNA levels were quantified using the $2^{-\Delta\Delta CT}$ method as previously described[74].

Furthermore, culture medium was collected after incubation in the absence or presence of Insl3 (100 ng/mL) to quantify the testicular release of 11-KT, the main androgen in fish[75], as described previously[25].

**Analysis of germ cell apoptosis by TUNEL**. To determine the incidence of apoptosis, paraffin embedded testis tissue from wild-type and *insl3* knockout males

was subjected to deoxynucleotidyl transferase-mediated dUTP nick-end labeling (TUNEL). First, testis tissue was briefly washed with PBS and subsequently fixed in 4% PBS-buffered paraformaldehyde (4 °C, overnight). After a 30 min wash with PBS, testis tissue was dehydrated and embedded in paraffin. Sections of 4 μm thickness were treated with permeabilization solution (20 μg/mL proteinase K [Sigma-Aldrich] in 10 mM Tris/HCl, pH 7.4) for 15 min at 37 °C. Finally, testis tissue was incubated with the TUNEL reaction mixture (In Situ Cell Death Detection Kit, Fluorescein; Roche) in the dark at 37 °C for 1 h. After washing twice in PBS, sections were counterstained with propidium iodide, mounted in Vecta-shield antifade mounting medium (Vector Laboratories) and examined by confocal laser scanning microscopy (Zeiss LSM 700). Negative and positive controls were included in each experimental set up.

Images from fluorescent stained sections were analyzed using a custom CellProfiler and Ilastik segmentation and quantification pipeline[76] with minor modifications. For the quantification of the TUNEL positive signal, captured image files were photographed at ×20 magnification and analyzed for the % fraction of cell nuclei stained for TUNEL (Fig. 3C). Illumination correction was first applied to images in order to correct uneven illumination using CellProfiler. Thereafter, pixel classification was performed on nuclei with Ilastik in order to distinguish background from foreground. Segmentation of nuclei was performed by the Identify Primary Objects module on the Ilastik probability maps using a manual threshold. Positive stained cells were segmented by applying a global minimum cross entropy threshold with the same module. The percentage of positive cells was then exported as the count of positive cells divided by total identified nuclei multiplied by 100.

**Statistics and reproducibility**. GraphPad Prism 9.0.0 package (GraphPad Software, Inc.) was used for statistical analysis. All datasets were tested for normal distribution using a Shapiro–Wilk normality test. In cases where all groups to be compared passed the normality test, significant differences between groups were identified using two-tailed Student's t-test (paired or unpaired, as appropriate) or one-way ANOVA followed by Tukey's test for multiple group comparisons. For datasets with no normal distribution, the non-parametric two-sided Mann–Whitney test was applied. Data are represented as mean ± SEM (*$p < 0.05$; **$p < 0.01$; ***$p < 0.001$; ns, no significant changes observed). The results shown in Figs. 5A–C and 6A, B are from representative testis tissue culture experiments, which were repeated 2–3 times (using 5–12 individuals each time).

**Reporting summary**. Further information on research design is available in the Nature Research Reporting Summary linked to this article.

## Data availability

The complete raw RNAseq data of the six samples sequenced in this study (three biological replicates per condition) have been deposited in the NCBI GEO database under the accession number GSE152038. Expression levels of selected genes in control, germ cell-depleted, and testes with recovering spermatogenesis[25] were retrieved using the GEO data set GSE116611. All data generated or analyzed during this study are included in this published article (and its Supplementary information files). The source data underlying the graphs are provided in Supplementary Data 1 and 2.

## Code availability

CRISPR/Cas9 targets for the zebrafish insl3 gene were selected using ZiFiT Targeter software (http://zifit.partners.org/ZiFiT). DAVID Bioinformatics Resources 6.7 (http://david.ncifcrf.gov/) was used to retrieve GO terms from the list of DEGs and exported as the input for each functional enrichment analysis. Functional enrichment analyses were carried out using a plugin available at http://www.baderlab.org/Software/EnrichmentMap/ for the Cytoscape network environment. The human Protein Atlas database was used to retrieve information on PPARG/Pparg expression in testis tissue (https://www.proteinatlas.org/ENSG00000132170-PPARG/tissue/testis).

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

## Acknowledgements

The authors thank Esther Hoekman, Daniëlle Janssen, and Henk Westland for maintaining the zebrafish stocks and Henk van de Kant (all from the Science Faculty, Utrecht University, The Netherlands) for technical support, and to Chris Coomans and Dorian Luijkx for pilot studies and a literature review, respectively, in partial fulfillment of their Master thesis work at Utrecht University. This study was co-funded by the Research Council of Norway BIOTEK2021/HAVBRUK program with the projects SALMAT (n° 226221) and SALMOSTERILE (n° 221648). The authors thank the financial support provided by the Brazilian Foundation CAPES (project n° BEX:9802/12-6) and the China Scholarship Council (CSC; grant n° 201706310069), for the scholarships awarded to L.H.C.A. and Y.T.Z., respectively.

## Author contributions

D.C., L.H.C.A., J.B. and R.W.S. designed the research; D.C., L.H.C.A., Y.T.Z., D.S., T.F., K.O.S., B.N., Y-C.C. and J.B. performed the research; W.G, Y-C.C., M.J.d.B. and J.L. contributed with animal samples; D.C., L.H.C.A., Y.T.Z., D.S., B.N., J.B. and R.W.S. analyzed the data; and D.C., L.H.C.A., J.B. and R.W.S. wrote the manuscript.

## Competing interests

The authors declare no competing interests.
