## [Peer Review File · Communications Biology]

Reviewers' comments:

Reviewer #1 (Remarks to the Author):

In this manuscript, the authors report that zebrafish *Insl3* stimulate germ cell differentiation by interacting with two *Rxfgp2* receptor paralogues expressed by Sertoli and myoid cells. This in turn results in reduction of germ cell apoptosis as well as increase proliferation and differentiation of Aund spermatogonia. They also show that *Insl3* effects are mediated by RA and *Pparg* signaling.

Overall, the study is well-researched with the findings novel and interesting for readers of Communications Biology.

The only suggestion I have is for the authors to consider measuring 11-KT level from whole fish to test their hypothesis of compensatory effect in 9 months-old *insl3*^{-/-} males and deterioration of this effect in 12 months-old mutant males.

Reviewer #2 (Remarks to the Author):

The manuscript "Insulin-like 3 affects zebrafish spermatogenic cells directly and via Sertoli cells" interrogates the role of *insl3* in spermatogenesis in zebrafish. The authors utilize multiple approaches to attain broader knowledge of how *insl3* signals spermatogonial regulation and its interactions with other pathways, particularly retinoic acid and *pparg* signaling. The manuscript expands the current knowledge of spermatogonial regulation, making important contributions to this field. Overall the conclusions made are supported by the data presented. Major findings in the paper are:

- Identification and investigation of likely testis *Insl3* receptors
- Elaboration of the role of *insl3* in spermatogonial regulation through mutant analysis
- Exploration of pathways that *insl3* may interact with
- Functional analysis how RA and *pparg* signaling interact with *insl3* in spermatogonial regulation

Specific Comments:

1. It would be great if the expression of *rxfp2a* and *rxfp2b* genes could be confirmed by ISH or another method of endogenous detection. It is possible that some of the spotty expression is due to incomplete enhancer elements in the transgenic fish. However, the authors do address this in the discussion and state that ISH experiments were attempted but unsuccessful. I don't suggest that any more needs to be done for this manuscript but the authors could consider trying fluorescent in situ hybridization in the future or they could try to make a endogenous transcriptional reporter using recent methods for *crispr/Cas* driven insertions (e.g. Kimura et al 2014, DOI: 10.1038/srep06545)
2. In the tunnel assay the authors state that the primary TUNEL positive cells were spermatocytes and spermatids. This does look like the case in the 12 month old fish, but 9 month old fish look to have mainly somatic cells and maybe spermatogonia that are positive. The distinction should be made in reporting these results.
3. Please include more information about the two *ppar* mutations. How do each of these effect the protein? It would be helpful if a figure could be included describing these mutations similar to the one for the *insl3* mutations.
4. The differences in phenotypes of the *ppar* mutant phenotypes are not discussed. Could there be genetic compensation that might be different in the two alleles or background mutations that are causing differences? One way to test if there is another mutation in the background affecting the phenotype is to look at the phenotype of the transheterozygous fish. If the phenotype resembles only

one of the two mutant phenotypes then outcrossing of the outlier can help to get cleaner background and consistent phenotypes. We have had this experience with *sa* alleles that have different variations on the phenotype but affect the protein in similar ways. I don't suggest that the authors do this experiment for this manuscript but it would be worthwhile if they continue to study these mutations.

5. What is a possible explanation for the increase in *pparg* expression in older fish when it is decreased in the younger fish?
6. Please include the precise promoter fragments used to make the two transgenic lines in the methods – include primers that were used to amplify these from the genome.
7. Please include the allele designation for the *insl3* mutation.
8. Figure 1: In the figure legend please fully write out confocal laser microscopy since I don't believe the CLSM abbreviation has been used in the manuscript. For the zoomed in images in panel C, please include which stage is being shown in each panel in the figure legend
9. Supp Fig 2: The white letters on the panels are difficult to see
10. Fig 4: I'm unclear how to interpret the key in panel C. How do the differently shaded gray circles correspond to the figure?
11. Fig 7: nice summary image

Reviewers' comments:

Reviewer #1 (Remarks to the Author):

In this manuscript, the authors report that zebrafish *Insl3* stimulate germ cell differentiation by interacting with two *Rxfgp2* receptor paralogues expressed by Sertoli and myoid cells. This in turn results in reduction of germ cell apoptosis as well as increase proliferation and differentiation of Aund spermatogonia. They also show that *Insl3* effects are mediated by RA and *Pparg* signaling.

Overall, the study is well-researched with the findings novel and interesting for readers of Communications Biology.

The only suggestion I have is for the authors to consider measuring 11-KT level from whole fish to test their hypothesis of compensatory effect in 9 months-old *insl3*^{-/-} males and deterioration of this effect in 12 months-old mutant males.

*Response: We thank the reviewer for the positive and constructive comments. The capacity in the zebrafish facility and the wet labs will remain limited for some time (COVID-related restrictions regarding the presence of technicians/researchers). In this context, we decided – in consultation with the Utrecht University Committee for Experimental Animal Care and Welfare – to keep, if at all possible, mutant lines showing the potential for animal welfare problems, as heterozygous lines. Older (starting at ~15 months of age) homozygous *insl3* KO mutants can develop skeletal deformities, so that this line was included in the “keep as heterozygous” approach. Since we have no homozygous fish “swimming” at present, we would have to wait for about 1 year to generate the samples and then quantify 11-KT levels, i.e. we are not able to follow the suggestion at this point in time. However, we have edited the Discussion (see lines 327-330 of the revised manuscript), and propose to quantify, in future experiments, plasma and testicular androgen levels. Quantifying whole body steroid levels has been used successfully in zebrafish research recently (e.g. <https://doi.org/10.1530/JOE-20-0160>) to characterize the loss of function phenotype of steroidogenic enzymes. However, in the present case, we suspect that quantifying blood plasma and testicular androgen levels may be more informative with respect to signal strength emanating from the testis and potential, intratesticular androgen insufficiency effects on spermatogenesis in mutants at 9 and 12 months of age.*

Reviewer #2 (Remarks to the Author):

The manuscript “Insulin-like 3 affects zebrafish spermatogenic cells directly and via Sertoli cells” interrogates the role of *insl3* in spermatogenesis in zebrafish. The authors utilize multiple approaches to attain broader knowledge of how *insl3* signals spermatogonial regulation and its interactions with other pathways, particularly retinoic acid and *pparg* signaling. The manuscript expands the current knowledge of spermatogonial regulation, making important contributions to this field. Overall the conclusions made are supported by the data presented. Major findings in the paper are:

- Identification and investigation of likely testis *Insl3* receptors
- Elaboration of the role of *insl3* in spermatogonial regulation through mutant analysis

- Exploration of pathways that insl3 may interact with
- Functional analysis how RA and pparg signaling interact with insl3 in spermatogonial regulation

Specific Comments:

1. It would be great if the expression of rxfp2a and rxfp2b genes could be confirmed by ISH or another method of endogenous detection. It is possible that some of the spotty expression is due to incomplete enhancer elements in the transgenic fish. However, the authors do address this in the discussion and state that ISH experiments were attempted but unsuccessful. I don't suggest that any more needs to be done for this manuscript but the authors could consider trying fluorescent in situ hybridization in the future or they could try to make an endogenous transcriptional reporter using recent methods for crispr/Cas driven insertions (e.g. Kimura et al 2014, DOI: 10.1038/srep06545)

Response: We have edited the chapter "Effects of insl3 knockout". We agree with the reviewer that showing rxfp2a- and rxfp2b-driven transgene expression only is insufficient. Unfortunately, our ISH approaches were not successful. However, we now refer in this part of the Discussion also to the data shown in Fig. 1B: rxfp2b transcript levels remained unaffected following busulfan-induced germ cell depletion, suggesting somatic expression, while rxfp2a decreased strongly (and increased during germ cell recovery), suggesting germ cell expression. We think that this observation at least allows assigning rxfp2a expression to the germ cell compartment, and rxfp2b to the somatic compartment. As suggested by the reviewer, additional (more sensitive) methods for detecting endogenous gene expression will hopefully provide the information on the exact cellular sites of these receptor transcripts (see lines 306-310; revised version of the manuscript).

2. In the tunnel assay the authors state that the primary TUNEL positive cells were spermatocytes and spermatids. This does look like the case in the 12 month old fish, but 9 month old fish look to have mainly somatic cells and maybe spermatogonia that are positive. The distinction should be made in reporting these results.

Response: We agree with the reviewer and have included that in Results (lines 170-175 in the revised version) and in Figs. 3A and B.

3. Please include more information about the two pparg mutations. How do each of these effect the protein? It would be helpful if a figure could be included describing these mutations similar to the one for the insl3 mutations.

Response: As suggested, we have now added a schematic diagram (new Supplementary Fig. 5) showing the exon-intron structure of the pparg gene and the location of the two mutations. We also show the nucleotide sequences of the wild-type pparg gene versus the mutant sequences and the stop codons resulting from the mutations. Finally, we show the protein domains affected by the mutations, based on an alignment with the human protein.

4. The differences in phenotypes of the pparg mutant phenotypes are not discussed. Could there be genetic compensation that might be different in the two alleles or background mutations that are

causing differences? One way to test if there is another mutation in the background affecting the phenotype is to look at the phenotype of the transheterozygous fish. If the phenotype resembles only one of the two mutant phenotypes then outcrossing of the outlier can help to get cleaner background and consistent phenotypes. We have had this experience with *sa* alleles that have different variations on the phenotype but affect the protein in similar ways. I don't suggest that the authors do this experiment for this manuscript but it would be worthwhile if they continue to study these mutations.

Response: We thank the reviewer for the insightful comments and recommendations. Regarding background mutations, the work reported here was carried out with the F4 generation after three outcrosses against AB wild-type fish. We therefore assume that background mutations are no longer playing a prominent role. This has been mentioned in the revised version in the Methods section (lines 546-548).

*Regarding a discussion on the differences in phenotypes of the two *pparg* mutant alleles, we have now included additional information on the differences (see point above, in particular regarding the new Supplementary Fig. 5C). In the discussion, we have now pointed out the differences referring to the amino acid sequences. However, since there is no functional data published on the zebrafish mutant proteins, the degree of speculation based on sequence alignments is high. Therefore, the respective new section in the Discussion (lines 447-455) mentions potential differences in the capacity to bind to DNA and to perhaps function as dominant negative protein regarding *sa1737*, while *sa1220* probably can no longer bind to DNA at all and may be replaced by another *Ppar* family member, potentially explaining why its loss did not show a phenotype in our system.*

5. What is a possible explanation for the increase in *pparg* expression in older fish when it is decreased in the younger fish?

*Response: The possible explanation encompasses *pparg* and other signaling systems studied in the present work, and has been presented in the section of the Discussion entitled, "Effects of *insl3* knockout" (lines 333 and following; original version of the manuscript). There, we pose that up until 9 months of age, effects resulting from the loss of *Insl3* function are compensated (by unknown mechanisms) in part through up-regulation of androgen production and down-regulation of *pparg* expression. These compensatory mechanisms then apparently deteriorate since at 12 months of age, androgen and *pparg* signaling had flipped from higher than to lower than wild-type controls. This was accompanied by reduced activities of two other pro-differentiation signaling systems (retinoic acid, insulin-like growth factor 3). Jointly, these changes could be causative for the aggravated spermatogenesis phenotype observed in 12 months-old males. However, all this is hypothetical and requires additional work to provide a mechanistic understanding of the observations made.*

6. Please include the precise promoter fragments used to make the two transgenic lines in the methods – include primers that were used to amplify these from the genome.

Response: The information requested by the reviewer has been added in Methods (lines 515-520) and in the new Supplementary Table 2.

7. Please include the allele designation for the *insl3* mutation.

*Response: The ZFIN official allele designation for the *insl3* mutant reported in this study has been included in Results (line 126).*

8. Figure 1: In the figure legend please fully write out confocal laser microscopy since I don't believe the CLSM abbreviation has been used in the manuscript. For the zoomed in images in panel C, please include which stage is being shown in each panel in the figure legend

Response: Done.

9. Supp Fig 2: The white letters on the panels are difficult to see

Response: As suggested by the reviewer, we have replaced the white letters by black letters.

10. Fig 4: I'm unclear how to interpret the key in panel C. How do the differently shaded gray circles correspond to the figure?

Response: Gene Ontology enrichment significance is represented as a color gradient. The shaded gray circles indicate false discovery rate (FDR) adjusted p-value, from the most highly (0-5% FDR, dark gray shading; e.g. Biosynthetic process) to the less highly (10-15% FDR, light gray shading; e.g. Metabolic process) enriched Gene Ontology terms. Fig.4 legend has been adjusted accordingly.

11. Fig 7: nice summary image

Response: We appreciate reviewer's comment.

REVIEWERS' COMMENTS:

Reviewer #2 (Remarks to the Author):

The authors have adequately addressed all of my comments about the manuscript